# Preserving Authenticity in Urban Regeneration: A Framework for the New Definition from the Perspective of Multi-Subject Stakeholders—A Case Study of Nantou in Shenzhen, China

**DOI:** 10.3390/ijerph19159135

**Published:** 2022-07-26

**Authors:** Shuyang Li, Fei Qu

**Affiliations:** 1School of Architecture & Urban Planning, Shenzhen University, Shenzhen 518060, China; 2050321002@email.szu.edu.cn; 2Benyuan Design and Research Center, Shenzhen University, Shenzhen 518060, China; 3Shenzhen Key Laboratory of Architecture for Health & Well-Being (in Preparation), Shenzhen 518060, China

**Keywords:** urban regeneration, authenticity, multi-subject-centered, urban villages

## Abstract

Urban villages are a modern heritage in China that provide affordable housing for urban immigrants and accommodate diverse communities of cohabitation. The regeneration and displacement of urban villages in the past decade have raised the potential risk of social exclusion and led to debates regarding the preservation of cultural and social authenticity. This paper establishes a new conceptual framework for the definition of authenticity from multi-subject perspectives. Taking Nantou as a massive regeneration case, semi-structured interviews have been conducted with multiple stakeholders, involving planning officials, developers, designers, researchers, merchandisers, curation NGOs, local residents, and tourists. Key factors and concepts related to the multi-subject evaluation of the planning decision and its effect on urban regeneration have been identified, using a grounded theory approach for interview analysis. A further coding of the data reveals four cognitive dimensions in the subjects’ expression of authenticity. The shift in the definition of urban authenticity implies that stakeholders might use different notions of authenticity in negotiations to resist or embrace urban interventions. The multivariant definition framework of authenticity can be adapted to guide future regeneration strategies, and would motivate the proliferation of urban preservation to take social and negotiable character into its definition.

## 1. Introduction

In the fast urbanization process, informal settlements as a supplement housing solution for new citizens have become a typical phenomenon in many cities of developing countries. To accommodate the huge number of rural-urban migrants in a short period, fast construction ignoring building regulations usually led to poor living conditions. The regeneration approach of these disadvantaged neighborhoods remains controversial, as physical removal or commodification might collide with the community’s rights in the city [1]. In high-consumption cities, affordable housing is a minimum requirement for promoting mental and social well-being. Social isolation due to relocation can become a more severe health threat than poor sanitary conditions [2].

In China, since the Reform and Opening Up in 1979, the urbanization rate has developed from 17.9% to 64.7% in 2020 [3]. Due to migrant congregation and squatting in the city [4], “urban villages”, which is also known as *chengzhongcun* in Chinese, occurs in many fast urbanized cities to accommodate the increasing number of migrant workers [5]. In the past decade, many urban villages, such as Dachong and Caiwuwei in Shenzhen, have been demolished, which has led to an important debate regarding the preservation of authentic urban heritage from human-centered perspectives.

Urban villages are cultural relics shaped by diverse communities that structure a city’s social landscape. Surrounded by the rapid expansion of the city, these vulnerable communities form a unique social order and strong community ties [6], which have stimulated discussions on the preservation of socio-cultural authenticity in the process of urban regeneration. However, it is difficult to apply traditional evaluation of the effectiveness throughout regeneration, centered on historical heritage preservation, to the diverse and complex contexts that urban villages represent, incorporating a broad range of stakeholders and management issues [7]. The first research objective is to find the key issues and concepts underlying the evaluation of urban village regeneration.

In addition, previous studies have presented a lack of consistency in the definition and criteria of authenticity, leaving human-centered preservation in a difficult position. The notions of authenticity vary for different actors of urban development—developers, residents, tourists—who might employ authenticity as a tool to fulfill the demands of prioritized groups. The absence of a clear definition of authenticity has been argued to jeopardize urban heritage [8]. As authenticity is intertwined with a discourse of rights and power [9,10], the definition of authenticity from a historic dimension needs to be linked to the contemporary notion of publicness and the participation of multi-stakeholders. The second research objective is to find the defining factors of urban authenticity in a multi-subject perspective with possible reasons driving different understandings.

Therefore, this study will develop a new conceptual framework for the defining factors of authenticity in a multi-subject-centered approach. It discovers people’s evaluations of regeneration and their references to different dimensions of authenticity in the case study of Nantou Old Town in Shenzhen, China. The area was selected due to its incorporation with a broad range of stakeholders and leaving with debates regarding authenticity conservation. Semi-structured interviews have been conducted with 27 representatives of stakeholders, including planning officials, developers, researchers, NGOs, designers, residents, and tourists. Using the grounded theory approach for transcript analysis, a conceptual framework will be established based on coding and conceptualizing perceived regeneration effectiveness and authenticity by practitioners and the users. The paper will conclude with the difference between stakeholders regarding their understanding and adoption of authenticity discourses to addressing interventions.

It is particularly pertinent and fundamental to investigate the multi-subject understanding of authenticity in the massive regeneration of urban villages, as the incorporation of multi-subjects would motivate the proliferation of the authenticity notions [11], generating a new framework of approaches that take into account the human-centered and negotiable character of its definition. The framework can be adapted to different contexts to assess the prevalence of multi-subject-centered authenticity and detect situations where it is dismissed.

### 1.1. The Formation and Development of Urban Villages in Shenzhen

For many centuries, China was a rural society. After 20 years People’s Commune period, the Reform and Open Policy launched on a journey of fast urbanization. With the expansion of the urban area, many villages were included inside the urban territory. In this process, to avoid the compensation with high cost for both money and time, the government tends to expropriate the farmland instead of the built-up residential land of the villagers, thus consequent to the village settlement surrounded by urbanized area. While inside the village, the land use has been densified to get more rent from the new-coming rural migrant. Gradually such villages are spatially encompassed or annexed by new urban development, leading to the high-density formation of urban villages [5], as shown in Figure 1. This phenomenon first appeared in southeast coastal cities such as Guangzhou and Shenzhen in the 1980s and soon spread all over the fast urbanized cities in China.

### 1.2. Debates on the Regeneration: Demolish or Preservation?

For a long time, urban villages were considered a negative phenomenon with poor living conditions as regards their unregulated constructions. With the physical environment being characterized as “dirty”, “chaotic”, and “deteriorated”, urban villages were perceived as a fertile ground for crime and fake products [12]. Considering the poor physical environment in an urban village, demolishing is the best choice for the developers in most regeneration projects, with many urban villages rebuilt into high-rises or gentrified tourist attractions. Figure 2 shows many urban villages in Shenzhen, while Figure 3 shows the demolished and regenerated urban villages in the Shenzhen Special Economic Zone (SEZ, the core area of Shenzhen) and plans for the next 5–10 years.

While in recent research, urban villages are considered a cultural heritage that play an important role in the urbanization process. As a neighborhood characterized by a mixture of rural and urban society, urban villages provide not only affordable housing for the new citizens but also walkable street and vital life, different from the super blocks in the new urban development. Therefore, the policies of urban village regeneration have shifted to urban village renovation without demolition. The government proposed the so-called “comprehensive improvement” for improving the living environment of an urban village with gentle renovations [13], under the pressure of increasing compensation fees for land expropriation and public opposition arising from the destruction of the socio-cultural authenticity of urban villages.

In the context of a city-wide promotion of urban village regeneration, how to regenerate the urban village without destroying the memory and authenticity is a question urgent to discuss, which calls for an evaluation of the value of authentic urban heritage and the community-engaged regeneration approach.

### 1.3. The Evolution of Urban Authenticity: From Object- to Multi-Subject-Centered Definition

The notion of authenticity has evolved from an objective concept to complex social construction, integrating objective and subjective aspects of urban reality [14]. In a series of theoretical discourses from a historic dimension, the orthodox ’object-centered‘ interpretations of authenticity rely on physical sources of evidence, such as urban heritage, architecture, and exterior appearance [15].

When the historical value is considered a preferred strength for urban redevelopment, a ’value-centered‘ and ’experience-based‘ understandings of authenticity have been proposed [16], to attract tourists, the creative class, and future economic investment. Both approaches have raised objections for not bringing the visions of all possible stakeholders into the discussion and potential subsequent processes of gentrification [17].

Therefore, the definition of urban authenticity remains loosely formulated, which might not incorporate the complex stakeholders and management issues in urban villages. A multi-subject-centered definition of authenticity will be examined in the expressions of diverse stakeholders using interviews.

## 2. Materials and Methods

### 2.1. Grounded Theory (GT)

The paper analyses the concepts underlying individual opinions using a grounded theory approach and categorizes the concepts into different dimensions of authenticity. GT is a systematic human-centered research method applied to qualitative research. First introduced by Barney Glaser and Anselm Strauss in 1967 [18], it provides an in-depth analysis of interviews and observations and is widely used in research on acoustic perception, public health conditions, human-based environmental and urban studies [19,20,21]. In contrast with the hypothetico-deductive model in traditional scientific research, it inducts the theory from data to concepts and categories. GT is more suitable for vague social problems and complex situations, and is insufficient to incept hypotheses.

In this study, grounded theory is adopted to analyze the stakeholder’s understanding of the regeneration process in Nantou, identifying the factors and concepts related to regeneration effects and the value of urban authenticity. Data are collected from observations and in-depth interviews with 27 people involved in the urban regeneration process or who currently work or live in the village.

### 2.2. Study Site: Nantou

Nantou is a typical urban village in Shenzhen with 30,000 inhabitants, and 90% are a floating population without hukou [22]. Nantou is considered to be a historical and cultural landmark with a history dating back to 1700 years ago. However, the ancient town has gradually vanished during constant expansion [23], forming an urban village in the city center with complicated layering and patterns (see Figure 4). Since it was recognized as a historical site by the government in 2002, preservation plans were proposed as a prelude to urban regeneration. While until 2011, only a few pseudo-classic constructions and transformations were built, allowing Nantou to continue offering low-cost housing for the floating workers. In 2017, the 8th UABB (Bi-City Biennale of Urbanism/Architecture) was held in Nantou with the theme of “Cities, Grow in Difference”. Exhibitions and events were made to better understand the value of urban villages and scenarios for urban symbiosis. After the UABB, renovations proposed by the Urban Research Institute of Vanke were meant to accelerate Nantou’s transformation into a cultural and creative hub, with all buildings along the main streets being renovated. Until now, the regeneration is still undergoing. Figure 5 shows the map of the area with regenerated buildings, shaping a cross area linking historical architectures such as Southern Gate, Baode Temple, and Xinan County Administrative Building. While for the rest of the Nantou, the regeneration has not yet been spread to the whole.

### 2.3. Research Participants

According to GT, the selection of interviewees was mainly based on theoretical purpose and relevance [18]. At the beginning of the study, purposive sampling was used to select the research object that could provide the most information for the research question. In the later stage of the study, theoretical sampling was selected. Based on the coding and analysis of certain interview data, relevant concepts, and categories were determined on a theoretical sampling basis, and interviewees who could provide dense and rich information concerning the study were selected for interview.

To understand the perspectives concerning urban regeneration, street observations and interviews regarding regenerated and unregenerated areas were carried out as the first step. The interviewees included tourists, residents, store owners, and young people working in the renovated office buildings. In total, 18 people were interviewed on the street.

In the second step, to better understand the purpose and process of urban regeneration, nine practitioners involved in Nantou’s regeneration were interviewed by appointment. Five individuals worked in different departments of the developer company, with diverse working scopes including merchants, operations, property management, social studies, and executive director of the company. Four representatives of on-site NGOs, curators, architects, and landscape designers were also interviewed.

### 2.4. Data Collection: Observations and Interviews

A free-flowing interview framework with open-ended questions was used to collect stakeholders’ thoughts and activities related to the regeneration process. The interviews with nine practitioners sought to capture the following themes: (1) self-introduction, starting date, and reasons for working in Nantou; (2) responsibility and decisions made in the regeneration process of Nantou; (3) corporative departments in the regeneration process and their works; (4) key issues that have been paid most attention to in the process; (5) description of the environment before and after the regeneration; (6) opinion to the effectiveness of regeneration; (7) expected development of Nantou in the future. The interviews were undertaken one by one in a conference room at the interviewee’s company, taking place between May and June 2022. Each interview started with introducing the research background without mentioning the core concept of authenticity.

The street interviews with 18 site users were mainly focused on (4)–(6) of the above themes to address key issues of concern and personal evaluation of the effectiveness of regeneration. In addition, more than 4000 photographs were taken during the observations to examine the actual street behavior of the renewed and unrenewed streets, which were used as empirical evidence of the objective effectiveness of the regeneration. Statistics on the quantity of interview data are shown in Table 1.

### 2.5. Data Analysis

Based on the grounded theory, four steps were applied to the data analysis. By using Atlas.ti software, the transcripts were coded for emerging concepts and key issues related to the perceived effects of urban regeneration. Then the relationships of the concepts were categorized and linked to form the theory related to the core concepts. Examples of the coding process are listed in Table 2.

#### 2.5.1. Open Coding

In this step, the interview transcripts were coded sentence by sentence and paragraph by paragraph to let the questions emerge from the memos. The interviewee’s responsibilities, priorities, and evaluations in the regeneration process were extracted for identification.

#### 2.5.2. Identifying the Concepts

In this step, theoretical sampling was used to identify the concepts across multi-subjective evaluation on the regenerating effects of Nantou. Secondary questions were asked to develop strategy-oriented concepts that address the most concerned and challenging issues in the regeneration process. Linkages were made between the concepts to find the intrinsic logic between perceived effects and strategic concepts.

#### 2.5.3. Generating the Categories and Subcategories

In this step, concepts with connection and similarity were grouped to generate the categories and subcategories in the regeneration strategy from the perspective of all stakeholders. Key concepts in each category and the frequency of their occurrence in the transcript were provided.

#### 2.5.4. Discovery of the Core Category of “Authenticity” and Generating the Theory

With the identification of the linkages, the core category of “authenticity” was formed. An explanatory theory can be generated from the core category and the framework of defining factors related to authenticity. The network of the core concepts from the perspective of multi-subjects was generated.

## 3. Results

### 3.1. Concepts Related to the Effectiveness of Urban Regeneration

In the case of Nantou, 3 key issues and 10 categories of concepts are considered the most important effects of urban regeneration as perceived by the interviewees. Table 3 shows the framework of concepts with the number of frequencies each mentioned.

In the regeneration process of Nantou, the improvement of the physical environment is the most cited issue by the interviewees. Nantou before regeneration was associated with descriptions such as extremely high density, drainage problems, tangled wires, and broken pavements. For the developers and government, infrastructure construction is the most effective way to improve the living quality and attract more visitors. While for the Nantou inhabitant, the regeneration of infrastructures is remarkable progress. The streets are cleaner, the drainage system functions better, the unsightly wires overhead are less dangerous. The appearance of densely built morphology and diverse functions were also mentioned in high frequencies, as they preserve the texture of the urban village in terms of the material environment.

The conflict between public rights and private benefits is another key issue in the regeneration process. Even though the developers try to set up their business model without disturbing the life of the original inhabitants, they still have to raise the rent and introduce shops with high price products to cover the massive construction and operating costs. The interests and welfare of both tenants and business owners were mentioned as factors to be considered in the regeneration process. To balance public and private rights, NGOs are involved to better understand the needs of all the related subjects and build an inclusive community. More public spaces were designed for all showing a kind of inclusiveness in the urban village. Even though the NGOs tried to reduce the conflict between the original residents and the newcomers by providing more public functions such as museums, the gentrification process is irreversible.

Cultural Authenticity is a core issue stemming from Nantou’s identity of urban heritage. Even though the developers acknowledged the value of urban villages as cultural heritage, they have been unable to clarify how to renovate this area in a way that would not destroy the sense of locality and history authenticity. During the previous regeneration process, many historical architectures were rebuilt, and fake façades were made to show a sense of history. These might become the best footnote of so-called “creative destruction” [1]. Even though most interviewees mentioned the importance of authenticity, different participants showed various perspectives. The tourists give a sense of constructed historical landscape. While for the residents, the authenticity is to keep the ordinary life. The developers and designers show a kind of in-between concept of locality.

### 3.2. Conceptual Framework of Authenticity

In addition to the socio-economic effectiveness of urban regeneration, the preservation of authenticity was the most frequently mentioned issue in the interviews. In the description of Nantou pre- and post-regeneration (Q5), the respondents shifted the dimension of evaluation, which might imply an expanded concept of authenticity as urban regeneration progresses towards capital-led customization. The interviewees’ opinions related to authenticity in four phases of regeneration have been coded (Figure 6). Some interviews reveal the socio-economic issues in the regeneration process with its causes and consequences (Q4). These subjective attribution factors are also provided in the figure to fulfill a network of discourse.

In describing the authenticity of Nantou before its renewal, the dilapidation of the physical environment was highlighted with objective factors such as weak history, dense population, and low rents as main constraints.

Regarding the main concerns during regeneration (2017–2020), infrastructure upgrading and protective regeneration were the main concepts. In the descriptions of this period, more human-centered concepts emerged, including handling public conflicts and negotiating top-to-bottom communication. From a subjective point of view, the power structure ranging from the government to the developer was the reason for a strategic preference towards commercial operations in regeneration. At the same time, the weak foundation of urban village deliberations was considered to have led to the bottom-up strategy being an idealized but unattainable concept. At this stage, the involvement of multiple subjects and academic criticism had prevented the demolition of authentic cultural relics to a certain extent. Still, the pursuit of public-private profitability and the urgency of construction time were used as excuses for improper renewal.

As expected, gentrification, exclusion, and population displacement were inevitable topics when describing the state of the regenerated neighborhood (2020–2022). At the same time, the developers emphasized the interests of the newly arrived merchants. They tried to replace the inherent characteristics of local culture with the characteristics and localities of the introduced businesses, resulting in the metonymy of the concept of authenticity. It is worth emphasizing that Nantou’s landmark positioning set by the government as a city-wide cultural source also drove the value-oriented renewal decisions, yet this was inconsistent with Nantou’s authenticity as an ordinary urban village, allowing the authenticity of urban culture to replace the authenticity of village culture.

Eventually, the descriptions of the future by interviewees from different backgrounds showed a pluralistic symbiosis, embracing expectations for cultural values, economic balance, and resident autonomy. In their opinions, established business models and political intentions make it difficult to avoid value- and experience-driven approaches to future regeneration. However, the complexity of multiple subjects also might drive future regeneration to consider resident co-creation and more defined rights and responsibilities.

It is worth noting that the driving factors were also growing throughout the progress of regeneration, from the village’s self-condition to the city’s proposition. The involvement of multi-actors has prevented the destruction, but only to a certain extent, as developers have attached new quantity and temporal conditions to authenticity as compensation, such as only 20% non-profit business and motorcycle access after midnight.

To be more explicit about the shift in the definition of authenticity and the relationship between the concepts, the defining concepts of authenticity were further categorized into four core concepts—the ‘object-centered’, ’value-centered’, ’experience-based‘, and the newly identified ’multi-subject-centered‘ understanding of authenticity.

#### 3.2.1. The ‘Object-Centered’ Authenticity

The concept of authenticity is most commonly found in the description of material and physical conditions of Nantou, such as ‘traditional urban village’, ‘dilapidated building’, ‘high-density texture’, ‘poor living environment’, ’protective renovation’, ‘infrastructure upgrade’ and ’historical authenticity’. These concepts are mostly accompanied by statements of visual evidence that shows the consistency of history. The ‘object-centered’ notion has been most widely mentioned by the designers involved in the renovation of residential buildings.


*“The environment before the renovation was quite poor. We took it over and basically transformed it in a protective way, which is equivalent to demolishing two buildings without moving the principle and the outline, combining them into one building, which has primarily the same outline but with a brand-new pattern” (transcript of the interview with an architect).*


In contrast, the discourse of the developer seems to deny the historical authenticity of Nantou, proposing that ancient buildings were fake and Nantou’s historical buildings had been destroyed in the previous generations of management instead of now.


*“Before the renewal, Nantou was just a normal urban village, with all the advantages and disadvantages of Shenzhen’s urban village. I do not think its history is true, to be honest. I think Shenzhen has no respect for history. If this place is properly protected, its authenticity will not be so false” (transcript of the interview with one of the Vanke developers).*


#### 3.2.2. The ’Value-Centered’ Authenticity

Among the interview with developers, the object-centered criteria of authenticity that rely on a true source of information was merely mentioned, replaced by a notion of regeneration taking into account the economic benefits, which is inconsistent with the definition of so-called ‘value-centered’ authenticity. Since the 1994 Nara Charter defined authenticity as the essential factor for attributing values [16], private developers might use the concept to present a top-down view that emphasizes ’economic development‘, ‘business model’, ‘the rights of the store owners’, ’benefit sharing‘, ‘strong connection with merchants’, ’commercial uniqueness‘ and so on.

In the interview, developers attempted to replace the subjects of local residents with new coming merchants. Much effort has been made to promote multiple sales paths with the aim of allowing merchants to survive in the post-epidemic period. In this way, the marketable value-centered business strategy has become a reasonable approach that would create an economically sustainable neighborhood.

#### 3.2.3. The ‘Experience-Based’ Authenticity

The ‘experience-based’ definition of authenticity stems from the tourism and the creative city ideal [14]. In the transcript, this is related to the concept of creating an image of ‘quality of life’, such as ‘de-scenarization’, ‘taste for young people’, ‘cultural label’, ‘enjoyable life’, ‘creative neighborhoods’, and ‘function replacement’. From the developer’s perspective, the feeling of conviviality—a status of cohabitation and interaction—could be created by a regular provision of public activities, festivals, and street curation events, even though they are far from the daily life of residents. The developer maintains the site’s image favored by the creative class through activity operations and property management, as reflected in the careful selection of non-chain stores and resistance to spontaneous trading. The label of the neighborhood’s image was erected from the first moment, which might be questionable in terms of inclusiveness and spatial justice [1].


*“I’m actually more concerned about the brand of Nantong Old Town, saying that the brand might be a little bit bigger, or what is our brand content? Of course, we may have a brand positioning, and then we have some research on the brand, but how the outside world perceives this is very important. For example, if we say we are a benchmark project for a more free, fresh, organic, and sustainable culture, then the clientele is most important for a project. If the clientele is not right, it means that the tenants may not fit in with our activities” (transcript of the interview with one of the Vanke developers in the Planning Department).*


#### 3.2.4. The ‘Multi-Subject-Centered’ Authenticity

Taking into account all the opinions of multiple stakeholders in this study, a new framework for understanding authenticity has been proposed. The new notion emphasizes the governance and time dimension of this issue, as Mitchell reminded us that “public space is always and inescapably a product of social negotiation and contest” [24]. The sub-concepts of multi-subject-centered authenticity include ‘public accessibility’, ’daily life’, ‘time-sharing management and transit space’, ‘electric vehicle management’, ‘localization’, ‘demands from all parties and conflict resolution’, and ‘balancing public and private property rights’.


*“It has a very large number of indigenous people, which is what we find attractive because it is not an empty art or cultural park. It has a localized thing. It even has its own community of old Nantou, WeChat group, and tenant’s group. It has a very dense local network, this is its great advantage. But it is true that in 2020, there were some communication conflicts in the middle because of the renewal, and the indigenous people have very strong opinions about the renovation. For example, we hope that some entrance space is given to the public space, which involves more precise property boundary demarcation. Most of the conflicts were handled quite cleverly, for example, through some other compensations, so that the balance between public and private owners could have a better final handshake” (transcript of the interview with a leader of an NGO).*


### 3.3. Differences in Perceived Authenticity between Stakeholders

The network of concepts related to authenticity mentioned by different stakeholders is established (Figure 7). Different perspectives have been found between tourists and residents, developers and NGOs, and so on. The findings also proved the co-existence of at least three categories of authenticity definition among the stakeholders of the study site.

#### 3.3.1. Object-Centered Historical Authenticity in the Perspective of Tourists

As the most important town in the history of east Canton and the origin of Hongkong, Nantou is an old town with 1700 years of history. However, in the last decades, history was forgotten and new buildings with more floors were constructed, ignoring urban planning and regulation. As outsiders, tourists seek a cultural heritage far from their daily life. In this sense, even though there are only a few traditional architectural constructions left, a kind of cultural landscape faking the missing details is indeed provided by the developers and designers to attract tourists, thus leading to a kind of constructed authenticity.

#### 3.3.2. Value-Centered Authenticity/Locality in the Perspective of Developer and Designers

When talking about the renovation of buildings in Nantou, the developers and designers emphasized the sense of locality in the urban village with local stores, products, and food. Many local brands were invited to have a shop in the regenerated area of the urban village, thus promoting the local design in Shenzhen. However, this type of locality is not rooted in Nantou, which is newly cultivated to meet the preferences of the creative class and treated as a symbolic icon for the tourist, as criticized by the researchers.

#### 3.3.3. Multi-Subject-Centered Authenticity in the Perspective of Residents and NGOs

For residents and visitors, the most fascinating feature of an urban village is the vivid daily life. Contrasting life between cities and villages, residents in the urban village have a more traditional relationship. However, many residents mentioned that they seldom go to the renovated area. On Zhongshan East Street, a formerly bustling commercial street, the daily market and tailor stores have been replaced by new businesses such as coffee shops, bookstores and museums. Even though some family restaurants are preserved, the price doubled due to the increased rent. While gentrification has been accepted as the price that must be paid for urban regeneration, residents and autonomous organizations call for a degree of public participation, either through time-phased management or the division of responsibility, so that real urban life from the perspective of underprivileged groups can be seen.

## 4. Discussion

Since the 2000s, the phenomenon of the urban village has attracted the attention of socialists and urban planners [6,23]. Previous studies mainly focused on the emergence process and morphology of urban villages at an urban scale. Few demonstrated the human-centered approach that linked authenticity to the confrontation of diverse stakeholders.

This paper demonstrates the understanding of authenticity from in-depth interviews with 27 stakeholders involved in the urban regeneration of Nantou, showing the conflict between developers, tourists, and residents. The findings confirm the hypothesis that urban regeneration has multiple attributes and effects in different dimensions. The study reveals the effects of urban regeneration in the three dimensions of physical space, social equity, and cultural authenticity with respective strategy-oriented concepts, using Nantou as an example. The second hypothesis is also confirmed by the fact that different actors employ distinct definitions when evaluating the preservation of authenticity. The emerging four categories of ideologies, from object-centered logic to value- and experience-centered approaches, correspond well to the evolving authenticity definitions in the previous literature on urban heritage conservation [14,25]. A trend has been shown regarding the understanding and tolerance of authenticity from visual-evidence-based evaluation to socio-economic forced demanding. This is also inconsistent with the aim of urban regeneration in flux, from physical improvement to economic development and cultural renaissance.

The shift in the definition of authenticity may be due to the dominant role of various organizations at different stages of regeneration. Our results suggest that the political demands of the government and the determination of a single developer have detached the concept of authenticity from the historical and human dimensions of the village to incorporate the needs of urban-scale development and attract new consumption boomers. It is important to note that the advocation for a value-centered authenticity and creating the experience of consumer groups might create a ‘staged authenticity’ or authenticity based on the tourists’ tastes [14], which could lead to homogeneity [26].

The new dimension of the ‘multi-subject-centered’ approach of authenticity responses to the community-engaged method of contemporary urban regeneration [27]. It shows the concept that has been dismissed in the former interpretations of authenticity by adding a time-sharing and negotiated manner of urban governance. Compared with the street observation on the main and 2 side streets in Nantou, the behavior at different times of the day shows what Bork et al. call the ‘transient urban space’—the openness and fluidity of space. The appropriation of the main street at night and sustaining everydayness on side streets can be seen as compensation and a negotiated mechanism in Nantou’s regeneration.

The importance of this paper lies in the intersection of spatial, historical, economic, and cultural demands in urban regeneration presented from a multi-subject perspective. The results of the paper will help identify the needs of different stakeholders as well as help future regeneration initiatives explore strategic models for weighing multiple interests. However, the study still has some shortcomings, such as the causes for the varying definitions arising from the interviewee’s subjective explanations. Future studies could combine in-depth interviews and objective data to explore the impact of the differences in the understanding of urban authenticity on the effectiveness of completed regeneration.

## 5. Conclusions

The rapid urbanization of Chinese cities has resulted in thoughts on urban regeneration mainly derived from practical experience, lacking the guidance of theoretical frameworks. In the last decade, the focus on preserving the historical and cultural values of urban villages has led to discussions on authenticity. This paper investigates the understanding of authenticity preservation by various stakeholders in the urban regeneration of Nantou through interviews and field observations. Three key issues and ten categories of concepts are identified to present the perceived impacts of urban regeneration by different stakeholders. The plurality of intersubjective views is also reflected in the articulation of authenticity. This paper summarizes the key concepts of authenticity in four critical dimensions of thinking, from object-centered to value- and multi-subject-centered approaches, forming a framework of authenticity centered on different claims of interest. Using discourse analysis, the paper shows different viewpoints between stakeholders. While the planer and developers are constantly pursuing industry upgrading and commercial uniqueness for young consumers, the researchers and residents call for a low cost of living based on an authentic population. The shift in the definition of authenticity might be attributed to the growing ambition of stakeholders involved, which expands the concept of the authenticity of a village to the authentic culture of a city. These differences in understanding of authenticity might imply the adoption of authenticity as an ideology discourse to negotiate and address interventions.

The proposed framework of authenticity from multi-subject perspectives and in different dimensions can be used to assess the preservation of multi-subject-centered authenticity in different contexts. The results would motivate the proliferation of authenticity preservation that considers human-centered and negotiable character in its definition. The framework has set a base for future studies that can apply quantitative methods to assess the contribution of each concept, with the aim of generating evaluation criteria for urban authenticity preservation.

## Figures and Tables

**Figure 1 ijerph-19-09135-f001:**
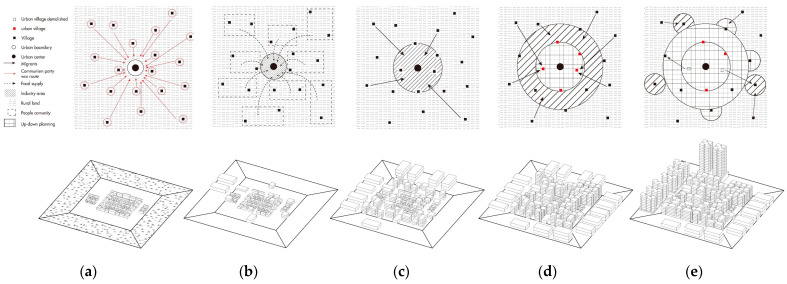
The Development Process of Urban Village: (**a**) Traditional village: city scale and block scale; (**b**) Village of People’s Commune: city scale and block scale; (**c**) Village included inside the city: city scale and block scale; (**d**) Densified Village: city scale and block scale; (**e**) Intensified Village: city scale and block scale.

**Figure 2 ijerph-19-09135-f002:**
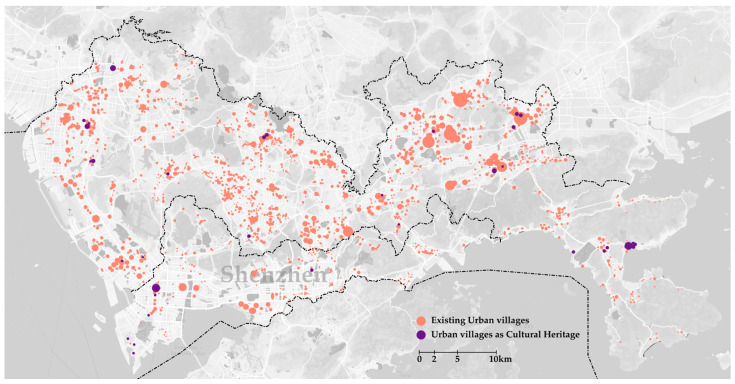
Existing urban villages in Shenzhen: data from Shenzhen Urban Planning Bureau.

**Figure 3 ijerph-19-09135-f003:**
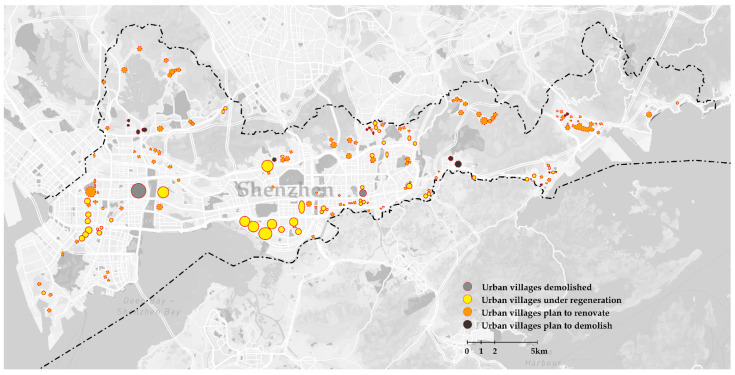
Development program of urban villages in the core area of Shenzhen.

**Figure 4 ijerph-19-09135-f004:**
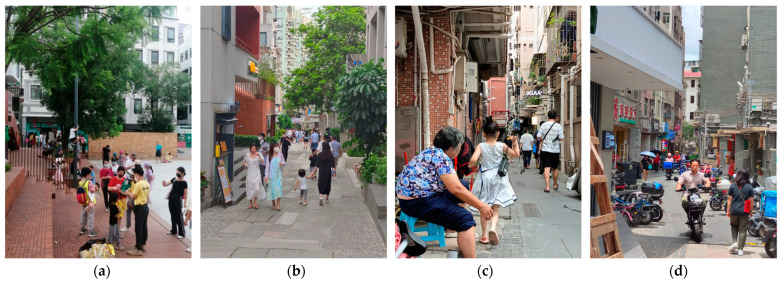
Nantou Old Town: (**a**) Regenerated area: Baode Square; (**b**) Generated area: Zhongshan South Street; (**c**) Unregenerated area; (**d**) Unregenerated area.

**Figure 5 ijerph-19-09135-f005:**
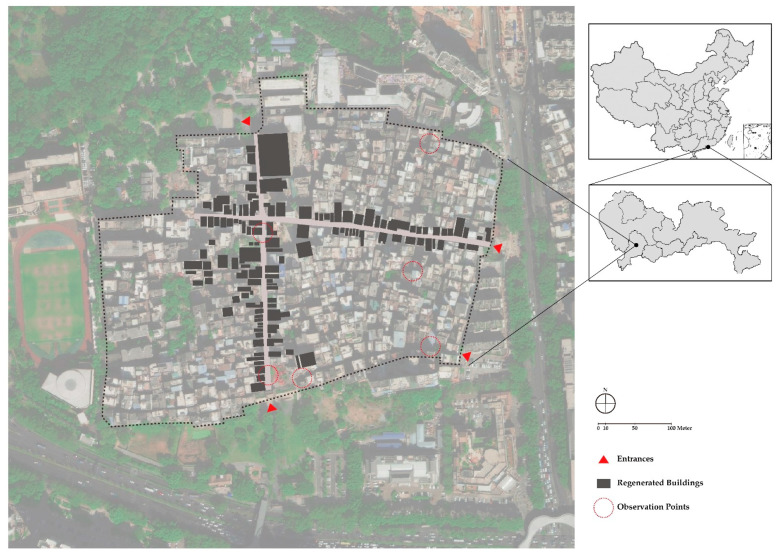
Study Area: Nantou Old Town.

**Figure 6 ijerph-19-09135-f006:**
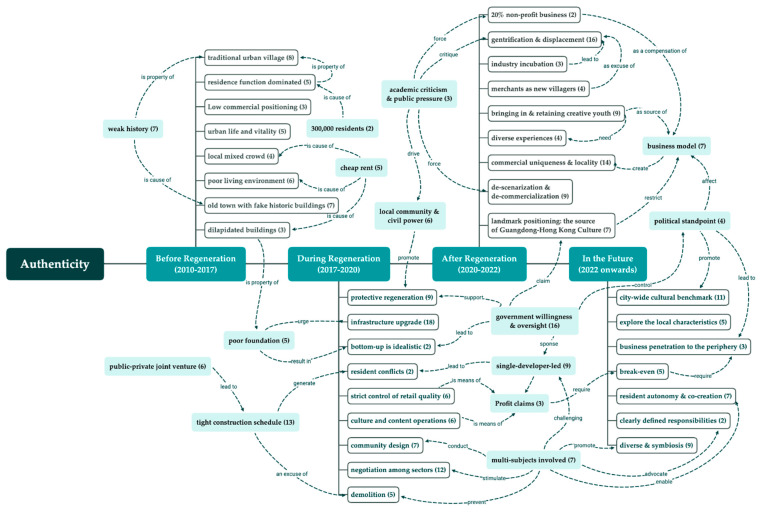
Codes related to authenticity in describing different regeneration phases with attributing factors and term frequencies.

**Figure 7 ijerph-19-09135-f007:**
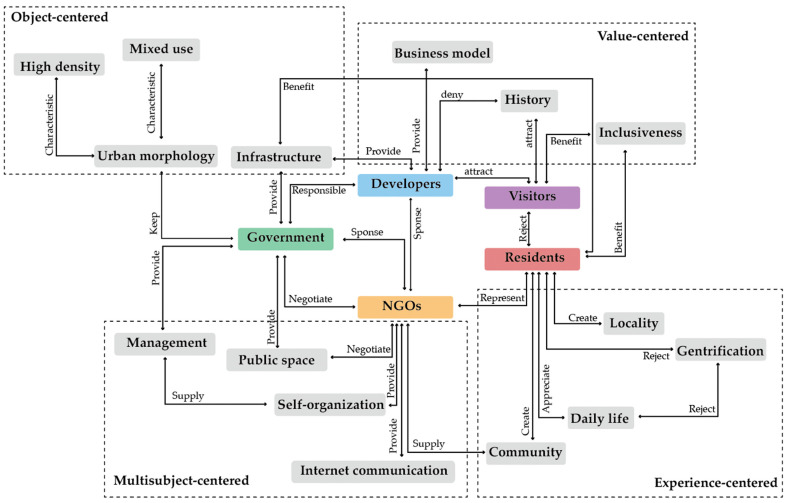
Network of core concepts and their relationship provided by groups of interviewees.

**Table 1 ijerph-19-09135-t001:** Participants and quantity of the interview data.

Participants	Number of Participants	Word Courts of Transcripts							
		Q1 Introduction	Q2 Responsibility	Q3 Corporation	Q4 Concern	Q5 Before & After	Q6 Effectiveness	Q7 Future	Total
Inhabitants	9	-	-	-	1982	753	1683	453	4871
Commuters	5	-	473	-	447	586	822	510	2838
Tourists	4	-	-	-	183	153	384	683	1403
Developers	5	1945	3254	7854	10,307	2158	4244	4779	34,541
NGO	4	1437	3923	3001	11,061	2315	11,894	2875	36,506
Total	27	3382	7650	10,855	23,980	5965	19,027	9300	80,159

**Table 2 ijerph-19-09135-t002:** Coding process for open coding, axial coding, and selective coding based on GT.

Memos	Open Coding	Conceptualizing Data	Categorizing Data
“History: capital of eastern Guangdong, source of Guangdong and Hong Kong.” “I hope Nantou will become a pluralistic community with” “Make tenants and residents aware of the benefits of urban renewal” “The biggest problem is the small block texture of the village within the city” “Multiple experiences for visitors” “The regenerated Nantou is more diversified” “Diversified business types, diversified customer groups, and diversified architectural forms” “Community and street culture” “Space inclusive, pet-friendly, elderly, children, and other people friendly” “Upgrading of business mode due to the investment in transformation and cultural industrial zone planning” “We are in the process of investment, the first one to avoid it a scenic, no matter the format or brand above “I think both the strength and problem of Nantou is a large amount of aboriginal status, and that’s what we find very attractive about it” ……	v1 history of Nantou …… v3 diversity of the community v4 Benefits for all v5 the urban texture …… v10 diversity in experiments v11 diversity in visitors v4, v5 …… v22 keep the street culture v23 inclusiveness …… v46 investment and the profit model …… r23 Avoid commercialization …… n7 Show local characteristics	C1 history (v1, v2, v7, v49, d9, d8, r26, r29) C2 mixed-use (v3, v11, d14, d2) C3 urban morphology (v21, d4, n20, r9) C4 infrastructure (v6, v8, v19, v55, n22, r3, r5, r11) C5 business model (d17, v46) C6 high density (v5, r12) C7 inclusiveness (v10, v23, v28, d10, v4, v53, r14) C8 locality (v36, d1, d3, d12, d11, n7, n6) C9 gentrification (v54, d5, d7, n18, r16, r17, r18, r21, r23, r24) C10 daily life (v22, v47, n11, d6) C11 community (n8, n14, d15) C12 management (r2, r7, v17) C13 public space (v20, n9, d13, d16) C14 Self-organization (v30, v51, n23) C15 Internet Communication (r30, d7)	CC1 Physical environment CC1c1 urban morphology CC1c2 infrastructure CC1c3 high density CC2 Social equality CC2c1 public space CC2c2 inclusiveness CC2c3 business model CC2c4 gentrification CC3 Culture Authenticity CC3c1 locality CC3c2 history CC3c3 ordinariness
	134 items	15 items	3 categories & 10 subcategories

**Table 3 ijerph-19-09135-t003:** Key concepts in the multi-subject evaluation of the effectiveness of Nantou’s regeneration.

Categories	Subcategories	Related Concepts (Term Frequency)
Improvement of Physical Environment	Urban morphology	High-density texture (6), residential buildings with characteristics of different periods (3), public squares (3), main streets (15), back streets (10), Mixed-use (4)
	Infrastructure	Electricity supply system (3), drainage system (5), parking (2), gas pipe (1)
	Governance	Security (3), clean-keeping (3), emergency handling (2), pandemic control (1), electric vehicle management (3)
Establishment of Social Equality	Publicness	Public events (7), public life (4), public accessibility (7), balancing public and private property rights (4)
	Inclusiveness	Children friendly (1), pet friendly (1), benefits for all (4), co-living (5), vitality (7), demands of the store owners (5), benefits for the original inhabitants (11)
	Business model	Marketing (9), area efficiency (2), rent (5), business environment (5), branding (4)
	Gentrification	Replacement of inhabitants (10), function replacement (5), rising rents, consumption upgrading (8), taste for young people (7)
Preservation of Cultural Authenticity	Locality	Local brand (7), local design (6), local food (7), localization (7), bottom-up process (7), Self-organization (3), self-construction (4)
	History	History of Nantou (8), Traditional architecture (3), protective renovation (7), history authenticity (5), fake façade (3), cultural label (3)
	Ordinariness	daily life (4), vegetable market (2), space for strolling (6), community (3)

## Data Availability

The data presented in this study are available upon reasonable request from the corresponding author. The data are not publicly available in deference to participants, who were not informed that their replies, even when de-identified, would be made publicly available.

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
