# Peer review of "Preserving Authenticity in Urban Regeneration: A Framework for the New Definition from the Perspective of Multi-Subject Stakeholders—A Case Study of Nantou in Shenzhen, China"

_ijerph, 2022, doi:10.3390/ijerph19159135_

Round 1

Reviewer 1 Report

Part 1.2 Debates on the Regeneration: Demolish or Preservation? it is worth supplementing with figures, if any. How many such urban villages were demolished? When did the process come to a halt? How has this number changed since it slowed down, because the text shows that it has not stopped completely?

The table "Table 1. Type, Participants / Observation Points and Quantity of the data" presents data on the length of the interviews and the volume of transcription. They are not qualitative in nature. If they are to remain, they should be supplemented with important issues, such as the number of issues analyzed in relation to the issues included in the interview questionnaire, completeness of answers, assessment of objectivism. This would be an important introduction prior to analyzing encoded transcripts.

Both in the "3.2 Conceptual Categories Related to Authenticity" and "3.3 Differences in Perceived Authenticity between Stakeholders", only to a small extent the conclusions drawn from the interviews are illustrated, however the collected material allows for this in extenso. Moreover, there is the impression that the conclusions are insufficiently empowered if this illustration is missing from the quotations.

Reviewer 2 Report

The title and the abstract coincide with the content of the paper. Keywords are well-chosen.

In the abstract, I propose to emphasise the aim of the work, the research question posed or the hypothesis. The abstract must focus on objectives, mention how they were achieved, and emphasize the results obtained (in this abstract, the results of the study are presented correctly)

I propose in the introduction should specify the methodology of research and research hypotheses. The diagnosis itself should indicate the novelty of the results and to publish the considerations in scientific journals. It should define the purpose of the work and its significance, including specific hypotheses being tested. The current state of the research field should be reviewed carefully and key publications cited.

In the discussion chapter I propnuate to answer the questions:

What did we establish new in our research?

What did others know, and what do we know?

What are the similarities and differences in the results?

What conclusions can be drawn from this?

What research plans do we have?

Did our results confirm the hypothesis?

In this part of the paper, we should show what our results mean in general and why our analyses are important.

Conclusion is optional, you may want to combine this part with a discussion

The methodology is correctly explained.

The figures/tables are appropriate.  They are easy to interpret and understand.

All the cited references are relevant to the research.

Round 2

Reviewer 1 Report

Most of the earlier remarks were taken into account. The changes made to the text could also be reflected in Conclusions, as the arguments have been expanded and the Conclusions could thus better reflect the scientific value of the text.

Author Response

Thanks very much for your suggestion. We have revised the conclusions. We also take this opportunity to shorten the abstract and change the expression of some phrases.
